# Intrinsic sequence specificity of the Cas1 integrase directs new spacer acquisition

Clare Rollie[1], Stefanie Schneider[2], Anna Sophie Brinkmann[3], Edward L Bolt[3], Malcolm F White[1]*

[1]Biomedical Sciences Research Complex, University of St Andrews, St Andrews, United Kingdom; [2]Faculty of Medicine, Institute of Cell Biology, University of Duisburg-Essen, Essen, Germany; [3]School of Life Sciences, Queen's Medical Centre, University of Nottingham, Nottingham, United Kingdom

**Abstract** The adaptive prokaryotic immune system CRISPR-Cas provides RNA-mediated protection from invading genetic elements. The fundamental basis of the system is the ability to capture small pieces of foreign DNA for incorporation into the genome at the CRISPR locus, a process known as Adaptation, which is dependent on the Cas1 and Cas2 proteins. We demonstrate that Cas1 catalyses an efficient trans-esterification reaction on branched DNA substrates, which represents the reverse- or disintegration reaction. Cas1 from both *Escherichia coli* and *Sulfolobus solfataricus* display sequence specific activity, with a clear preference for the nucleotides flanking the integration site at the leader-repeat 1 boundary of the CRISPR locus. Cas2 is not required for this activity and does not influence the specificity. This suggests that the inherent sequence specificity of Cas1 is a major determinant of the adaptation process.

## Introduction

The CRISPR-Cas system is an adaptive immune system present in many archaeal and bacterial species. It provides immunity against viruses and other mobile genetic elements mediated through sequence homology-directed detection and destruction of foreign nucleic acid species (reviewed in *Sorek et al., 2013*; *Barrangou and Marraffini, 2014*; *van der Oost et al., 2014*). Organisms with an active CRISPR-Cas system encode one or more CRISPR loci in their genomes. These comprise a leader sequence followed by an array of short, direct, often palindromic repeats interspersed by short 'spacer' sequences, together with a variable number of CRISPR associated (*cas*) genes. Spacers are derived from mobile genetic elements and constitute a 'memory' of past infections. The CRISPR locus is transcribed from the leader, generating pre-CRISPR RNA (pre-crRNA) that is subsequently processed into unit length crRNA species and loaded into large ribonucleoprotein complexes. These complexes, known as 'Interference', 'Effector' or 'Surveillance' complexes, utilize crRNA to detect and degrade cognate invading genetic entities, providing immunity against previously encountered viruses and plasmids.

The process of foreign DNA capture and integration into the CRISPR locus is known as 'Acquisition' or 'Adaptation' (reviewed in *Fineran and Charpentier, 2012*; *Heler et al., 2014*). This process underpins the whole CRISPR-Cas system, but is the least well understood aspect. Fundamentally, acquisition can be separated into two steps: the capture of new DNA sequences from mobile genetic elements, followed by the integration of that DNA into the host genome. New spacers are incorporated proximal to the leader sequence and result in the duplication of the first repeat (*Goren et al., 2012*; *Yosef et al., 2012*). The leader sequence proximal to the repeat is important for integration, but transcription of the locus is not required (*Yosef et al., 2012*). New spacers are frequently incorporated in both possible orientations (*Shmakov et al., 2014*). The integration process in *Escherichia coli* results in staggered cleavage of the first CRISPR repeat, where the 3′-end of one

*For correspondence: mfw2@st-and.ac.uk

Competing interests: The authors declare that no competing interests exist.

**eLife digest** In most animals, the adaptive immune system creates specialized cells that adapt to efficiently fight off any viruses or other pathogens that have invaded. Bacteria (and another group of single-celled organisms called archaea) also have an adaptive immune system, known as CRISPR-Cas, that combats viral invaders. This system is based on sections of the microbes' DNA called CRISPRs, which contain repetitive DNA sequences that are separated by short segments of 'spacer' DNA. When a virus invades the cell, some viral DNA is incorporated into the CRISPR as a spacer. This process is known as adaptation. CRISPR-associated proteins (or 'Cas' proteins) then use this spacer to recognize and mount an attack on any matching invader DNA that is later encountered.

Exactly how a spacer is inserted into the correct position in the CRISPR array during adaptation remains poorly understood. However, it is known that two CRISPR proteins called Cas1 and Cas2 play essential roles in this process.

Rollie et al. took Cas1 proteins from a bacterial cell (*Escherichia coli*) and an archaeal species (*Sulfolobus solfataricus*) and added them to branched DNA structures in the laboratory. These experiments revealed that Cas1 from both organisms can break the DNA down into smaller pieces. Cas2, on the other hand, is not required for this process. This 'disintegration' reaction is the reverse process of the 'integration' step of adaptation where the CRISPR proteins insert the invader DNA into the CRISPR array.

Rollie et al. also found that the disintegration reaction performed by Cas1 takes place on specific DNA sequences, which are also the sites where Cas1 inserts the spacer DNA during adaptation. Therefore, by examining the disintegration reaction, many of the details of the integration step can be deduced.

Overall, Rollie et al. show that selection by Cas1 plays an important role in restricting the adaptation process to particular DNA sites. The next step will be to use the disintegration reaction to examine the DNA binding and manipulation steps performed by Cas1 as part of its role in the adaptation of the CRISPR system.

strand of the incoming DNA is joined to the end of the CRISPR repeat in a trans-esterification (TES) reaction, with another TES reaction occurring on the other strand (*Diez-Villasenor et al., 2013*) (*Figure 1*, numbered yellow arrows). Intermediates in this reaction have been observed in *E. coli*, and the sequence of the leader-repeat1 junction is important for the activity (*Arslan et al., 2014*). The order of the two integration steps shown in *Figure 1* is not yet known, although it has been suggested that the reaction on the minus strand (site 2) occurs first in *E. coli* (*Nuñez et al., 2015*). The sequence at site 2 is flanked by the end of the first repeat and the first spacer, and is therefore inherently less well conserved. In *E. coli*, the last nucleotide of the newly copied repeat is derived from the first nucleotide of the incoming spacer, which imposes further sequence requirements on that system (*Swarts et al., 2012*).

Although shown in *Figure 1* as fully double-stranded, the incoming spacer DNA could have a partially single-stranded character. Recent work by Sorek and colleagues has shown that the *E. coli* RecBCD helicase–nuclease complex, which processes DNA double-strand breaks for recombination and degrades foreign DNA, is implicated in the generation of viral DNA fragments captured by Cas1 and incorporated into the CRISPR locus as new spacers (*Levy et al., 2015*). This confirms previous observations linking Cas1 with RecBCD (*Babu et al., 2011*) and raises some intriguing questions as RecBCD generates ssDNA fragments asymmetrically, generating fragments tens to hundreds of nucleotides long from the 3′ terminated strand and much longer fragments from the 5′ terminated strand (reviewed in *Dillingham and Kowalczykowski, 2008*). The Cas4 enzyme, which is a 5′ to 3′ ssDNA exonuclease (*Zhang et al., 2012*; *Lemak et al., 2013*), may provide ssDNA fragments for Cas1 in systems lacking RecBCD. However, it is difficult to see how two integration reactions could occur without two 3′ hydroxyl termini (*Figure 1*) and half-site integration is not observed with a ssDNA substrate (*Nuñez et al., 2015*). Potentially, the ssDNA fragments generated by these nucleases may re-anneal and experience further processing to generate partially duplex molecules of defined size prior to integration by Cas1.

Adaptation requires the products of the *cas1* and *cas2* genes in vivo and these are the most universally conserved of all the *cas* genes (*Makarova et al., 2006*). Cas1 is a homodimeric enzyme

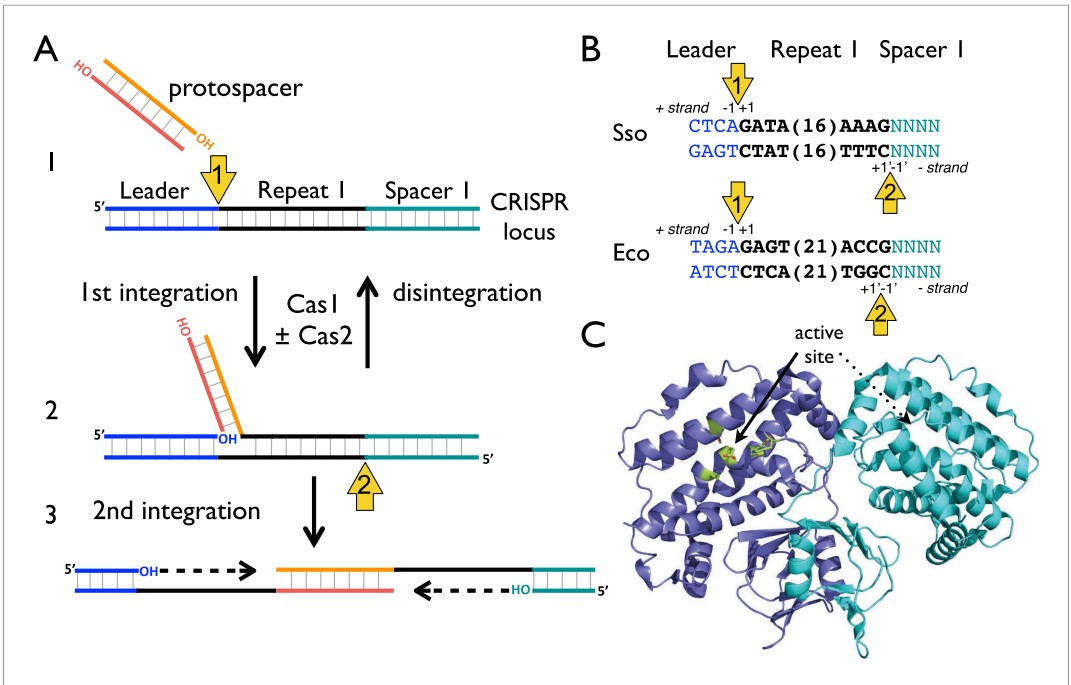

**Figure 1**. CRISPR spacer acquisition and Cas1. (**A**, 1) The 3'-end of an incoming protospacer attacks the chromosomal CRISPR locus at the boundary between the leader sequence and repeat 1. A trans-esterification (TES) reaction (yellow arrow 1) catalyzed by Cas1 joins the protospacer to the 5' end of repeat 1. For many integrases a (reverse) disintegration reaction can be observed in vitro. (2) Another TES reaction (yellow arrow 2) joins the other strand of the protospacer to the 5' end of repeat 1 on the bottom (minus) strand, resulting in the formation of a gapped DNA duplex. (3) The gapped duplex is repaired by the host cell DNA replication machinery, resulting in the addition of a new spacer at position 1 and replication of CRISPR repeat 1. (**B**) Sequences flanking the two TES reaction sites at repeat 1 in *Sulfolobus solfataricus* and *Escherichia coli* are shown. The leader is in blue, repeat in black and spacer 1 in teal. The number of central nucleotides of the repeat omitted from the sequence is shown in parentheses. (**C**) Structure of Cas1 from *Pyrococcus horikoshii* (PDB 4WJ0) with subunits coloured blue and cyan, showing the dimeric 'butterfly' conformation with the active site residues highlighted in green.

with a two-domain structure and a canonical metal dependent nuclease active site in the large domain formed by a cluster of highly conserved residues (*Wiedenheft et al., 2009*) (*Figure 1C*). *E. coli* Cas1 has nuclease activity in vitro, with activity observed against double- and single-stranded DNA and RNA substrates (*Babu et al., 2011*). Some specificity was observed for branched DNA substrates, in particular for DNA constructs resembling replication forks (*Babu et al., 2011*). Initial biochemical analyses of a panel of archaeal Cas2 enzymes revealed an endonucleolytic activity against ssRNA substrates that could be abrogated by mutation of conserved residues (*Beloglazova et al., 2008*). In contrast, Cas2 from *Bacillus halodurans* has been shown to be specific for cleavage of dsDNA substrates (*Nam et al., 2012*). Recently however, Doudna and colleagues demonstrated that *E. coli* Cas1 and Cas2 form a stable complex in vitro and that the 'active site' of Cas2 was not required for spacer acquisition, suggesting that Cas2 may not have a catalytic role in spacer acquisition in vivo (*Nuñez et al., 2014*). It is probable that Cas2 acts as an adaptor protein, either bringing two Cas1 dimers together or mediating interactions with other components necessary for spacer acquisition. Recently, Nunez et al. demonstrated that *E. coli* Cas1 can integrate a protospacer into a supercoiled plasmid in vitro, in a reaction stimulated by Cas2. Integration events were observed at the boundaries of most CRISPR repeats and in other areas of the DNA close to palindromic regions, implicating a role for palindromic DNA structure in the adaptation process (*Nuñez et al., 2015*).

In order to further mechanistic understanding of the spacer acquisition process, we have undertaken a systematic analysis of the underlying biochemistry. We demonstrate that Cas1 catalyses TES of branched DNA substrates efficiently in vitro in a reaction that represents the reverse- or disintegration of an incoming spacer from the CRISPR locus. The disintegration reactions catalysed by

diverse integrases have proven a powerful model system for the understanding of the mechanism of integration. For Cas1, the reaction is strongly sequence dependent with the specificity matching the predicted integration site 1 for both *E. coli* and *Sulfolobus solfataricus* Cas1, and does not require Cas2.

## Results

### Cas1 catalyzes a TES reaction on branched DNA substrates

The Cas1 and Cas2 proteins from *S. solfataricus* (CRISPR-Cas subtype IA) and *E. coli* (CRISPR-Cas subtype IE) were expressed in *E. coli* with N-terminal polyhistidine tags and purified to homogeneity by metal affinity and gel filtration chromatography. Previously, it was demonstrated that *E. coli* Cas1 (EcoCas1) cleaved branched DNA substrates preferentially (*Babu et al., 2011*). We investigated the activity of *S. solfataricus* Cas1 (SsoCas1) against a DNA substrate with a 5′-flap structure (*Figure 2*). By labelling the single-stranded 5′-flap of the downstream duplex at the 5′-end with radioactive phosphorus, we observed cleavage of the flap by SsoCas1, releasing an 18 nt product (*Figure 2B*). A variant of SsoCas1 where the active site metal ligand glutamic acid 142 was changed to an alanine (E142A) was inactive, implicating the canonical nuclease active site of Cas1 in the reaction. This result appeared consistent with the earlier studies for EcoCas1 (*Babu et al., 2011*) and suggested that the ssDNA flap was removed at or close to the branch point. The activity was independent of the presence or absence of SsoCas2, suggesting that Cas2 is not involved in this nuclease activity in vitro.

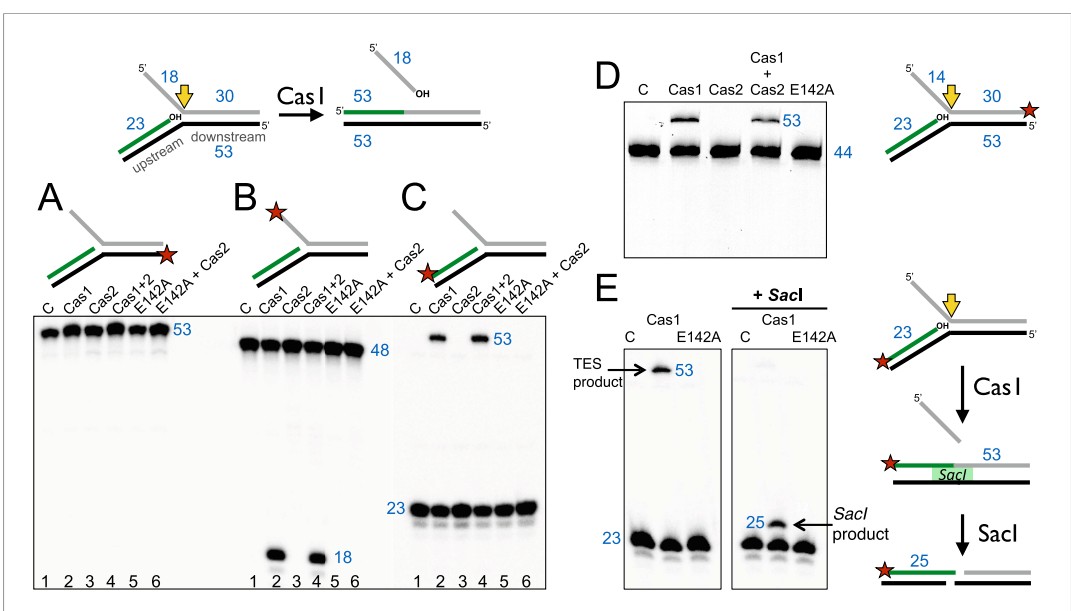

**Figure 2**. Disintegration of a branched DNA substrate by SsoCas1. Denaturing gel electrophoresis was used to analyse the products generated by SsoCas1 with a branched DNA substrate (Substrate 1). The 5′ flap (18 nt) was released when the phosphodiester backbone was attacked by the 3′-hydroxyl group at the branch point. The reaction required active Cas1 and was independent of Cas2. DNA lengths are shown in blue (nt). The TES site is indicated with a yellow arrow and the labelled strand with a red star. (**A**) Shows reactions with the continuous strand (black) labelled; (**B**) with the flap (grey) strand labelled and (**C**) with the upstream (green) strand labelled, each on the 5′ end. Lanes: 1, control with no added protein; 2, WT Cas1; 3, Cas2; 4, Cas1 + Cas2; 5, Cas1 E142A variant; 6, E142A Cas1 + Cas2. (**D**) The 5′-flap strand was labelled on the 3′ end with a fluorescein moiety, and the flap reduced to 14 nt (Substrate 1-FAM). Cas1 catalyses the TES reaction generating a 53 nt labelled strand. Lane C: control incubation in absence of Cas1. (**E**) TES reactions were carried out with SsoCas1, or the E142A active site mutant, on a fork substrate containing a nicked SacI restriction site spanning the branch point (SacI substrate). A TES product of 53 nucleotides is visible in lane 2 containing Cas1. The right-hand panel shows the effect of adding SacI restriction enzyme after the TES reaction. The TES product is no longer visible, but a shorter product of 25 nucleotides is present indicating regeneration of the SacI recognition sequence by the TES reaction.

We proceeded to label the other strands of the substrate to follow the reaction products in more detail. The continuous (black) strand was not cleaved by SsoCas1 (*Figure 2A*). However, when the 23 nt (green) strand of the upstream duplex presenting a 3′-hydroxyl end at the junction point was labelled we observed the formation of a new, larger DNA species (*Figure 2C*). This observation was consistent with a joining or TES of the upstream DNA to the downstream DNA strand by Cas1. By switching to a label at the 3′ end of the downstream duplex we confirmed that the reaction catalyzed by Cas1 involved the joining of the upstream and downstream DNA duplexes with concomitant loss of the 5′-flap (*Figure 2D*). The lack of evidence for any shorter DNA species in *Figure 2D* was consistent with the conclusion that we were observing a TES rather than a nuclease reaction. Again, the TES reaction was unaffected by the presence or absence of Cas2 and was dependent on the wild-type active site of Cas1, as the E142A variant was inactive.

In order to define the product of the TES reaction in more detail, we modified the sequence of the branch point to introduce an interrupted site for the restriction enzyme *SacI* across the junction. A precise TES reaction at the branch point to generate duplex DNA would result in the 'repair' of the restriction site, generating a substrate for *SacI*. SsoCas1 processed this substrate generating the 53 bp TES reaction product. On addition of the *SacI* restriction enzyme, the 53 bp species was converted to a 25 bp product (*Figure 2E*). This confirmed that the Cas1-mediated reaction resulted in the formation of a functional *SacI* site in vitro, consistent with a precise TES reaction at the branch point.

It is likely that the TES reaction catalyzed by Cas1 with branched DNA substrates in vitro represents the reverse or disintegration of an integration intermediate, as observed recently for EcoCas1 (*Nuñez et al., 2015*). We therefore tested EcoCas1 with the same set of branched substrates, showing that manganese, magnesium and cobalt all supported the same robust disintegration activity in the absence of Cas2, with no nuclease activity observed (*Figure 3A*). Given that Eco and SsoCas1 are divergent members of the Cas1 family, this suggests that the disintegration activity is likely to be a general property of Cas1 enzymes. Experiments where the concentration of Cas1 was titrated against a fixed concentration of substrate (50 nM), showed that maximal activity plateaued above 250 nM for EcoCas1 and was maximal from 100 to 500 nM for SsoCas1 (*Figure 3B,C*).

To characterise the specificity of the disintegration reaction in more detail, we examined SsoCas1 activity for a variety of substrates differing in the nature of the 5′-flap or duplex arm released by the reaction (*Figure 4*). SsoCas1 was indifferent to the presence of duplex or single-stranded DNA in the 5′-flap, processing a nicked 3-way junction with a similar efficiency to the 5′-flap substrate. The disintegration reaction required the presence of the 3′-hydroxyl moiety at the branch point as a three-way (or Y) junction was not a substrate for Cas1. To confirm this observation we studied a 5′-flap substrate with a phosphate moiety at the 3′ end of the upstream strand in place of a hydroxyl group. As expected, this substrate did not support disintegration activity for either Sso or EcoCas1, with no larger TES product detected (right hand lanes). Tellingly, neither enzyme cleaved the 5′-flap of this substrate to generate shorter products (left hand lanes), confirming that Cas1 catalyses a concerted TES reaction rather than a sequential 'cut and join' activity.

## Sequence specificity of the disintegration reaction

Disintegration reactions are commonly seen for integrases and transposases, and represent a very useful means to study the underlying integration mechanism (*Chow et al., 1992*; *Delelis et al., 2008*) as the local arrangement of the DNA in the integrase active site is typically the same for the two reactions (*Gerton et al., 1999*). One prediction of this hypothesis is that the disintegration reaction could demonstrate some sequence preference if integration, which must happen at a specific, defined site in the CRISPR locus, is partly driven by the sequence specificity of Cas1. We therefore designed a family of related disintegration substrates by systematically varying the nucleotides flanking the TES site and tested how efficiently Cas1 could disintegrate these substrates. Having demonstrated conclusively that we could follow the progress of the disintegration reaction by monitoring the liberation of a displaced DNA strand from a 5′-flap substrate, we used this assay for all subsequent investigations.

## The +1 position

We first tested the importance of the nucleotide acting as an acceptor for the attacking 3′-hydroxyl of the incoming DNA strand (the +1 position) by varying the nucleotide at this position in the model

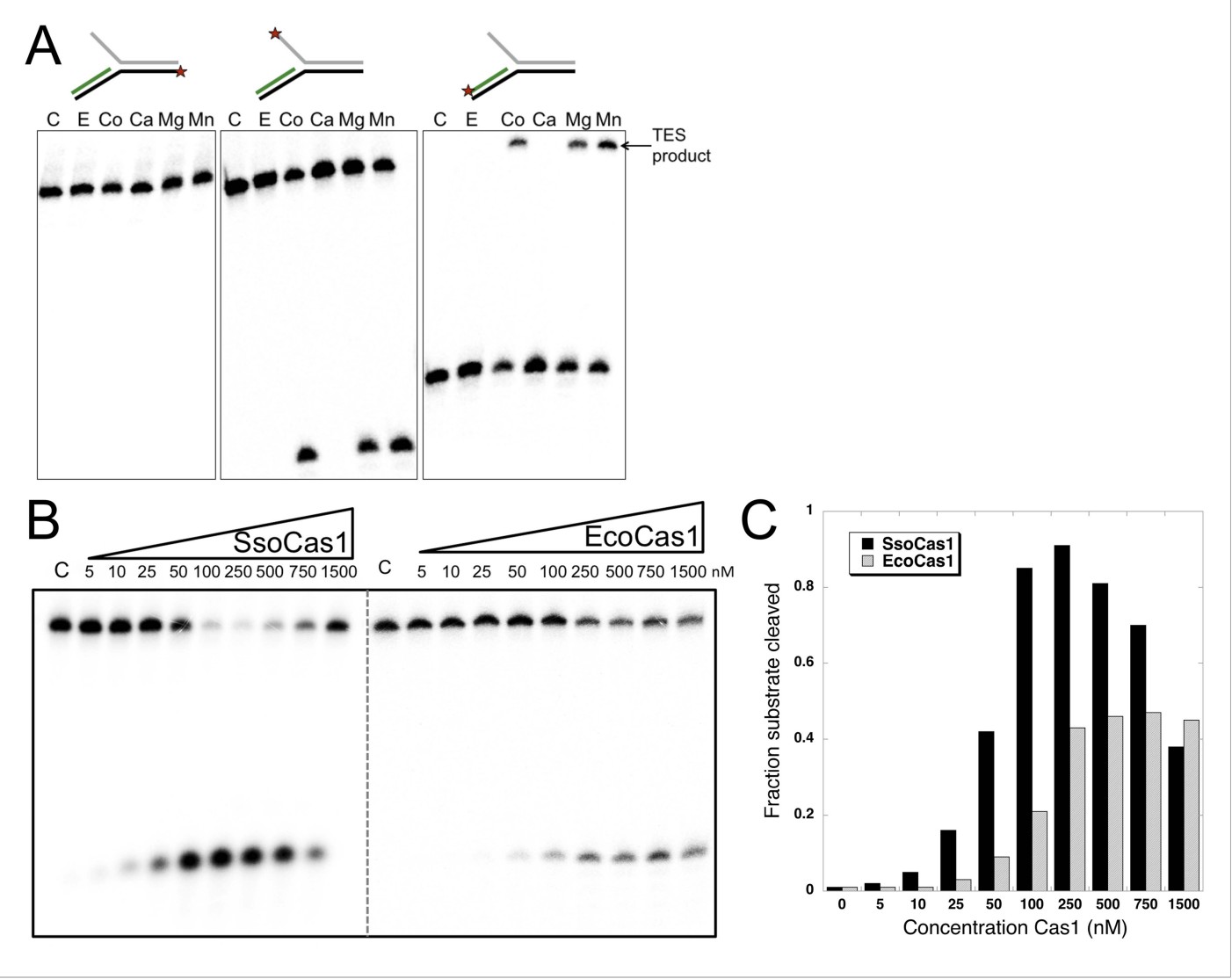

**Figure 3**. TES activity of *E. coli* and *S. solfataricus* Cas1. (**A**) *E. coli* Cas1 also catalyses an efficient metal dependent disintegration reaction. TES reactions were carried out under standard conditions, using Substrate 3 and varying the divalent metal ion as indicated. EcoCas1 showed robust TES activity in the presence of cobalt, magnesium and manganese. Each of the three strands of the substrate was labelled individually as for *Figure 2* (5′ label indicated by a star). Lanes were: c, substrate alone; substrate incubated with Cas1 and 5 mM of E, EDTA; Co, cobalt chloride; Ca, calcium chloride; Mg, magnesium chloride; Mn, manganese chloride for 30 min at 37°C. (**B**) Concentration dependence of Cas1 TES activity. Substrate 3 (50 nM) was incubated with the indicated concentration of Sso or EcoCas1 for 30 min under standard assay conditions and the reactants were analysed by denaturing gel electrophoresis and phosphorimaging. SsoCas1 showed maximal activity at 250 nM, representing a fivefold molar excess of enzyme over substrate, with a decline in activity above 500 nM enzyme. EcoCas1 had maximal activity that plateaued above 250 nM enzyme. (**C**) Quantification of the data (raw data provided in *Figure 3—source data 1*). These data are representative of duplicate experiments.

The following source data is available for figure 3:

**Source data 1**. Concentration Cas1.

substrates (*Figure 5*). In vivo, this acceptor nucleotide should represent the position in the CRISPR locus where new spacers are joined to the end of the repeat. The *S. solfataricus* CRISPR repeat has a guanine at one end and a cytosine at the other, each of which are predicted to act as acceptors for a new phosphodiester bond formed with the incoming spacer (*Figure 1B*). Consistent with this, we observed the most efficient disintegration activity with SsoCas1 when substrates had a guanine at the +1 position (*Figure 5A*). Assays were carried out in triplicate and the reaction products quantified,

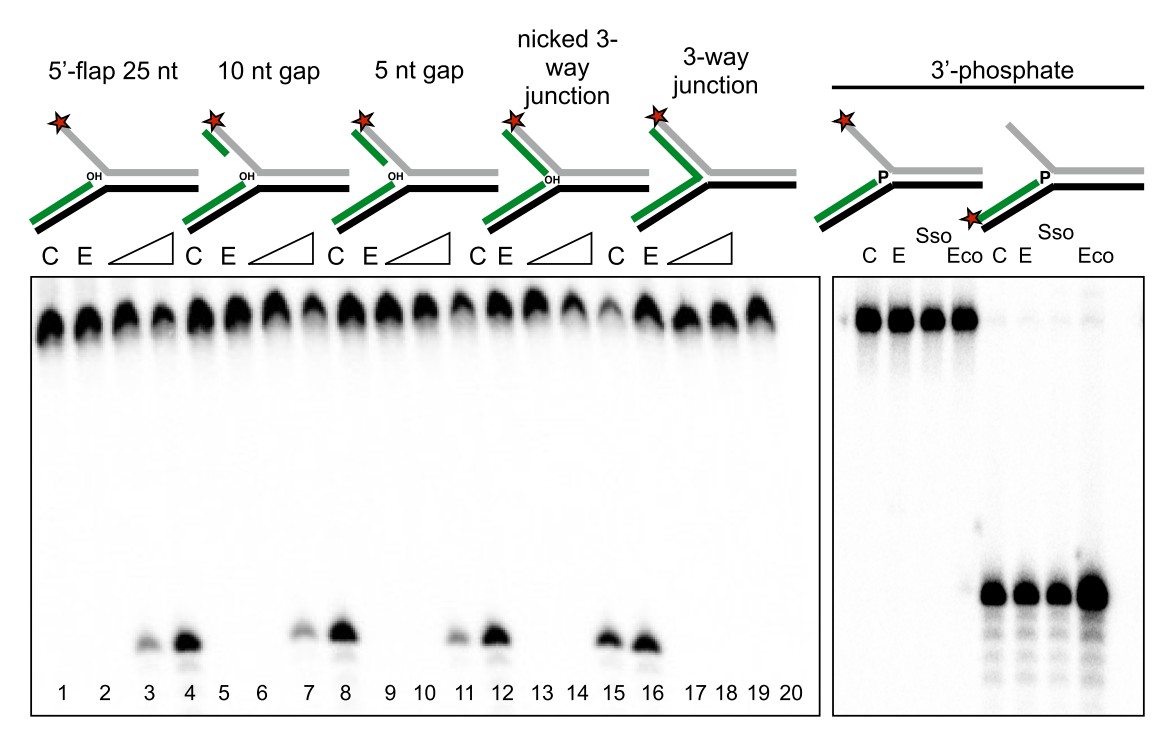

**Figure 4**. Importance of flap and 3′ terminus structure. The importance of the released 25 nt 5′-flap structure was investigated by varying the length of duplex DNA in that arm from 0 (canonical 5′-flap) to a full 25 bp (nicked 3-way junction) (left hand panel, all based on substrate 19). All supported robust disintegration activity by SsoCas1. An intact Y- junction did not support TES activity. Lanes: C, substrate alone (1, 5, 9, 13, 17); E, SsoCas1 E142A variant 30 min incubation (2, 6, 10, 14, 18); incubation with wild-type SsoCas1 for 10 and 30 min (other lanes). The right hand panel shows the effect of replacing the attacking 3′-hydroxyl moiety at the branch point with a phosphate group (3′ phos substrate) no TES or nuclease activity was observed for either Sso or EcoCas1. C, substrate alone; E, SsoCas1 E142A variant.

confirming the qualitative observation of a preference for guanine, followed by cytosine, adenine and thymine, with rates of 0.06, 0.013, 0.0058 and 0.0009 min$^{-1}$, respectively (*Figure 5B*).

For *E. coli*, the first nucleotide of the repeat is a guanine. Although the corresponding position at the other end of the repeat on the minus strand is a cytosine, it has been demonstrated that the new spacer joins at the penultimate residue, which is also a guanine (*Swarts et al., 2012*). EcoCas1 displayed a striking preference for a guanine at the +1 position for the disintegration reaction, with all three alternative nucleotides strongly disfavoured at this position (*Figure 5C*), in good agreement with the prediction based on the repeat sequence. For EcoCas1 the reaction did not go to completion and accordingly we fitted the data with a variable end-point as described in the 'Materials and methods' (*Figure 5D*). Although the reaction rates could not be determined accurately, guanine at position +1 supported rates at least 10-fold higher than any other nucleotide. We also tested the effect of inclusion of Cas2 on the sequence specificity of EcoCas1, and observed that Cas2 had no effect, with strong preference for a guanine at +1 still observed (*Figure 5E*).

## The −1 position

We proceeded to investigate the sequence dependence of the nucleotide at the 3′-end of the attacking DNA strand (the −1 position) in the disintegration reactions. For the first integration site, this should correspond to the last nucleotide of the leader sequence, which is an adenine in both *S. solfataricus* and *E. coli*. Although both Cas1 enzymes catalyzed the disintegration of substrates with an adenine at this position, clear sequence preference was not apparent (*Figure 6*). For *S. solfataricus*, the −1′ position on the minus strand is variable in sequence. However, in *E. coli*, site 2 occurs at the penultimate nucleotide of the repeat and therefore has a cytosine at the −1′ position (*Swarts et al., 2012*). In this situation, the incoming 3′ terminal nucleotide of the spacer has to be a cytosine in order

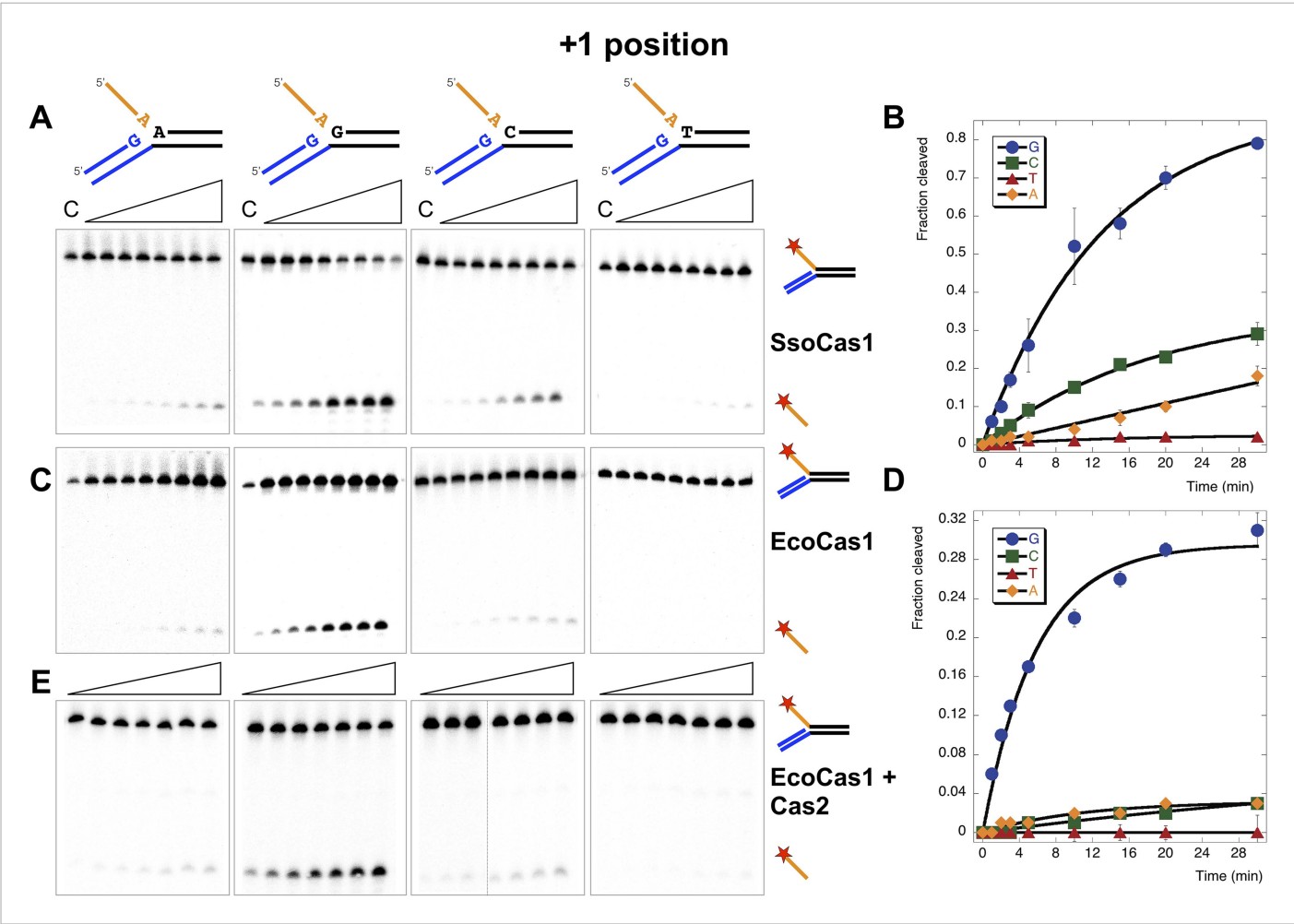

**Figure 5**. Sequence specificity of the disintegration reaction at the +1 position. The nucleotide at the acceptor (+1) position was varied systematically to assess the sequence dependence of the disintegration reaction carried out by Cas1 from *S. solfataricus* (**A**, **B**) and *E. coli* (**C**, **D**) (Substrates 3, 6, 7, 8). In the gels on the left (**A**, **C**) each substrate was incubated with Cas1 for 1, 2, 3, 5, 10, 15, 20 and 30 min in reaction buffer prior to electrophoresis to separate the cleaved 5′-flap from the intact substrate. C–control with no Cas1 added. The plots on the right (**B**, **D**) show quantification of these assays. Data points represent the means of triplicate experiments with standard errors shown (raw data provided in *Figure 5—source data 1* and *Figure 5—source data 2*). The data were fitted to an exponential equation, as described in the 'Materials and methods', and for EcoCas1 a variable floating end point was included to allow fitting as the reaction did not go to completion. The effect of Cas2 (150 nM) on EcoCas1 (150 nM) sequence specificity for substrates (50 nM) varying at position +1 (substrates 3, 6, 7, 8) was also tested (**E**). The second panel from the right is a composite image from two phosphorimages of the same time course as indicated by a black line.

The following source data are available for figure 5:

**Source data 1**. Nucleotide at +1 position.

**Source data 2**. Nucleotide at +1 position.

to complete the original repeat sequence. To mimic the TES substrate at this site more closely, we tested substrates where the first nucleotide of the 5′ flap, equivalent to the incoming nucleotide in the forward reaction, was a cytosine, but a cytosine at position −1 was still not favoured by EcoCas1 (*Figure 6C*). This may suggest that the disintegration reaction is not a good model for integration at site 2, which is further discussed later.

## The −2 position

The nucleotide at the −2 position, which is part of the conserved leader sequence for integration site 1, is also a potential determinant of integration specificity. Accordingly, we varied this residue

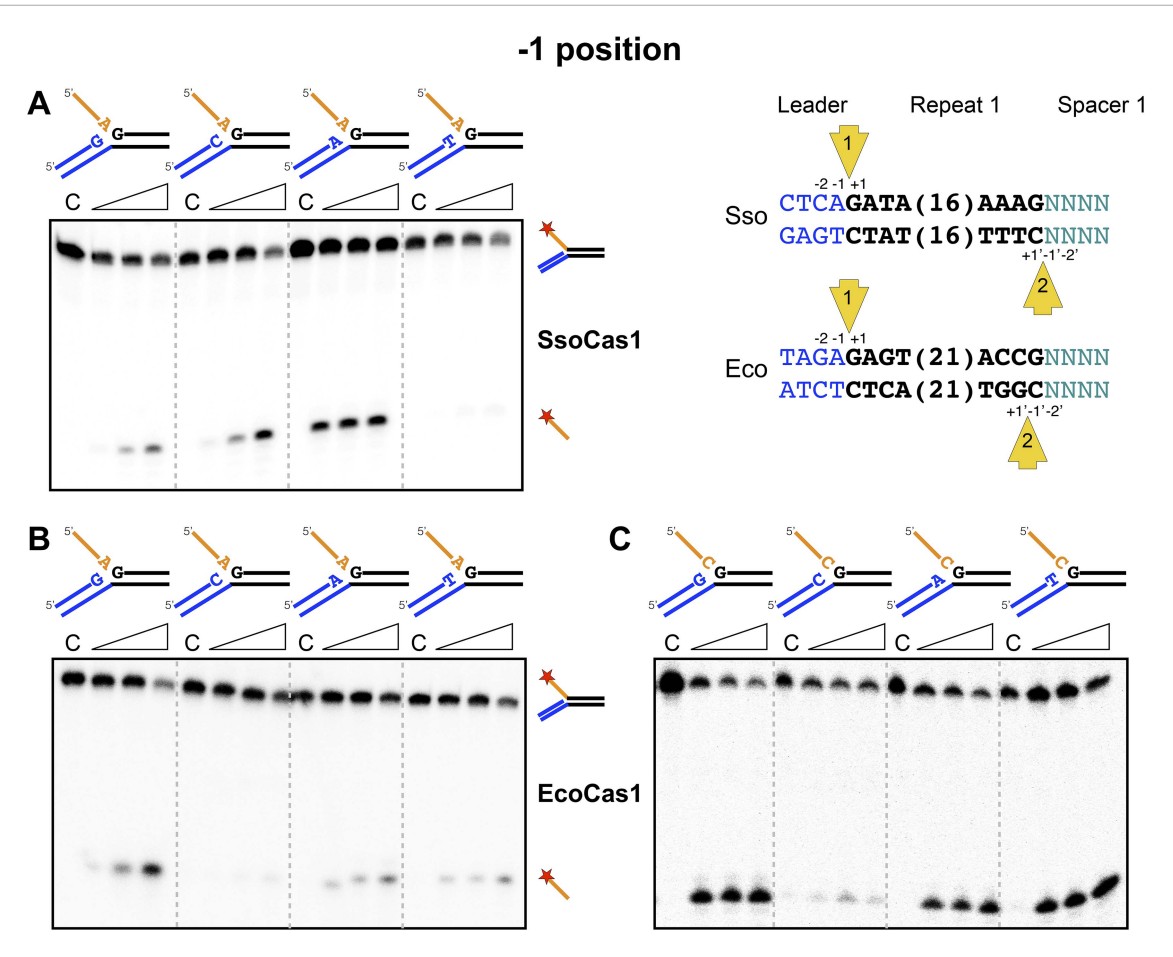

**Figure 6**. Sequence specificity of the disintegration reaction at the −1 position. The nucleotides participating in the disintegration reaction were varied systematically at the −1 position (substrates 3, 9, 10, 11). For SsoCas1 (**A**) there was some preference for adenine at this position, consistent with integration site 1. For EcoCas1 (**B**, **C**), a cytosine at position −1 was disfavoured over all other possibilities, even when the residue equivalent to the 'incoming' nucleotide was also a cytosine (substrates 15, 16, 17, 18). Each substrate was incubated with Cas1 for 5, 10 and 30 min in reaction buffer prior to electrophoresis. C–control with no Cas1 added.

systematically and investigated the efficiency of the disintegration reaction for both Cas1 enzymes (*Figure 7*). In *S. solfataricus*, the −2 position in the leader is a cytosine, which supported the strongest disintegration activity (*Figure 7A*). In *E. coli*, the −2 position in the leader is a guanine. A clear preference for guanine over all other nucleotides was observed for EcoCas1 (*Figure 7B*), confirmed by a more detailed kinetic analysis (*Figure 7C*) which was fitted as for *Figure 5D*. These data are consistent with a role for sequence discrimination at the −2 position by both enzymes, which is relevant for integration site 1 but not site 2, where this position varies depending on the sequence of the last spacer inserted.

## The incoming nucleotide

We next checked for specificity at the 3′ end of the 5′ flap in the disintegration product, which corresponds to the 3′ end of the incoming spacer during integration. No sequence preference was detected for SsoCas1 (data not shown), which is consistent with the essentially random nature of the incoming DNA. During adaptation in *E. coli*, the incoming nucleotide at integration site 1 is expected to be random, but at site two is always a cytosine, where it completes the repeat sequence (*Swarts et al., 2012*). For EcoCas1, an adenine or cytosine was strongly favoured over guanine and particularly thymine (*Figure 8*), suggesting discrimination by EcoCas1 at this position.

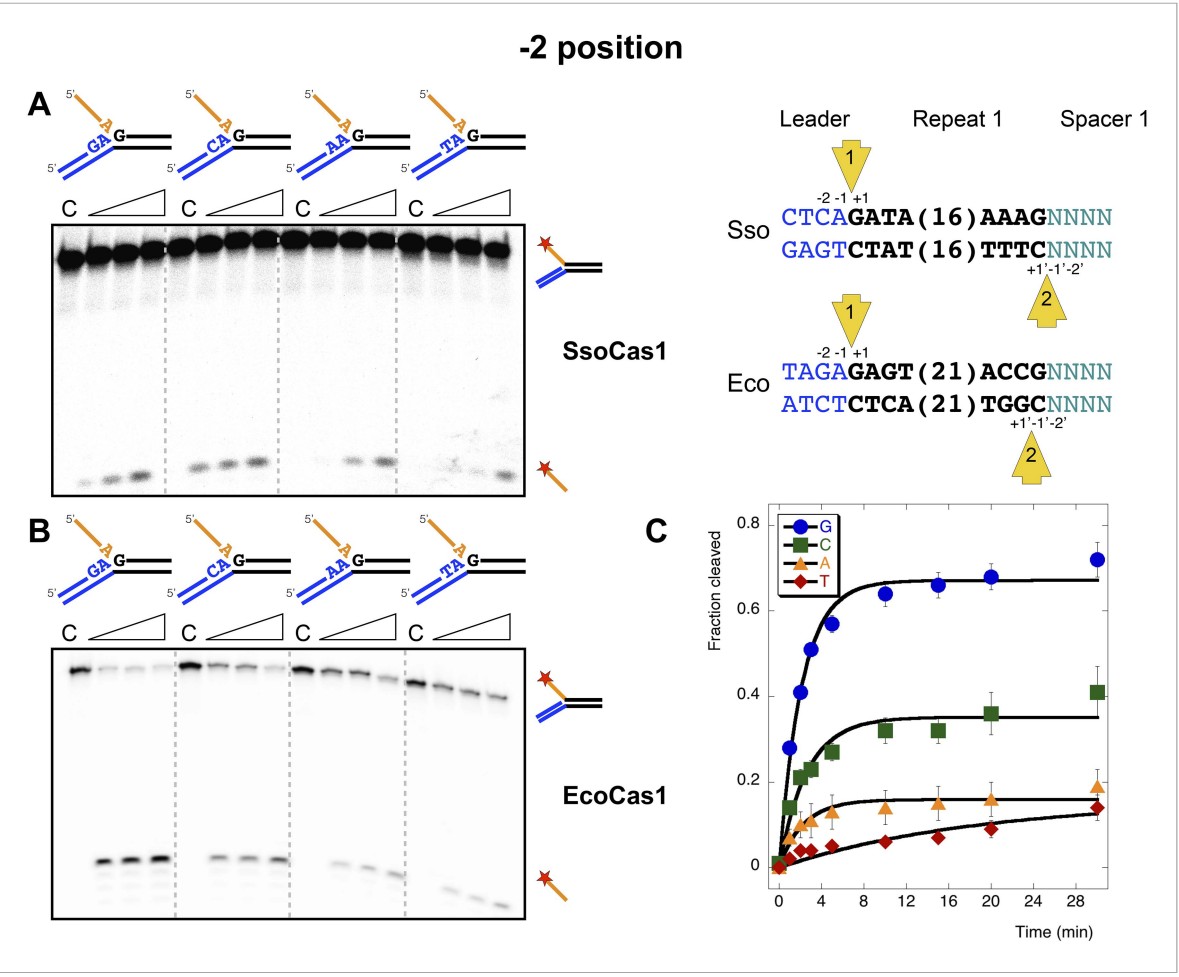

**Figure 7**. Sequence specificity of the disintegration reaction at the −2 position. The nucleotides participating in the disintegration reaction were varied systematically at the −2 position, which is a cytosine (Sso) or guanine (Eco) at integration site 1, and variable at integration site 2 (substrates 10, 12, 13, 14). (**A**) SsoCas1; (**B**) EcoCas1. Each substrate was incubated with Cas1 for 5, 10 and 30 min in reaction buffer prior to electrophoresis. C–control with no Cas1 added. (**C**) For EcoCas1, the clear preference for guanine at position −2 was confirmed by more detailed kinetic analysis (raw data provided in *Figure 7—source data 1*) as described for *Figure 5*.

The following source data is available for figure 7:

**Source data 1**. Nucleotide at −2 position.

## Disintegration of authentic *E. coli* integration intermediates

The substrates examined so far in this study do not correspond to the actual sequences encountered by EcoCas1 during integration. Accordingly, we constructed a pair of substrates that correspond to the products of integration when spacer 3 in the *E. coli* CRISPR array is integrated at site 1 (top strand) or site 2 (bottom strand) (*Figure 9*). These were constructed from oligonucleotides as before to generate disintegration substrates with a 5′ flap. Disintegration was analysed by denaturing gel electrophoresis, phosphorimaging and quantification of triplicate experiments. EcoCas1 disintegrated the site 1 substrate quickly, with the reaction reaching approximately 75% conversion in 3 min, which compares favorably with the best model sequence studied. Site 2 was also a good disintegration substrate, though it was converted significantly more slowly than site 1, perhaps due to the presence of a disfavored cytosine at position −1.

## Discussion

Studies of the CRISPR spacer acquisition process in vivo have yielded many key insights, but they are complicated by the fact that it is very difficult to separate the two distinct steps of spacer capture and

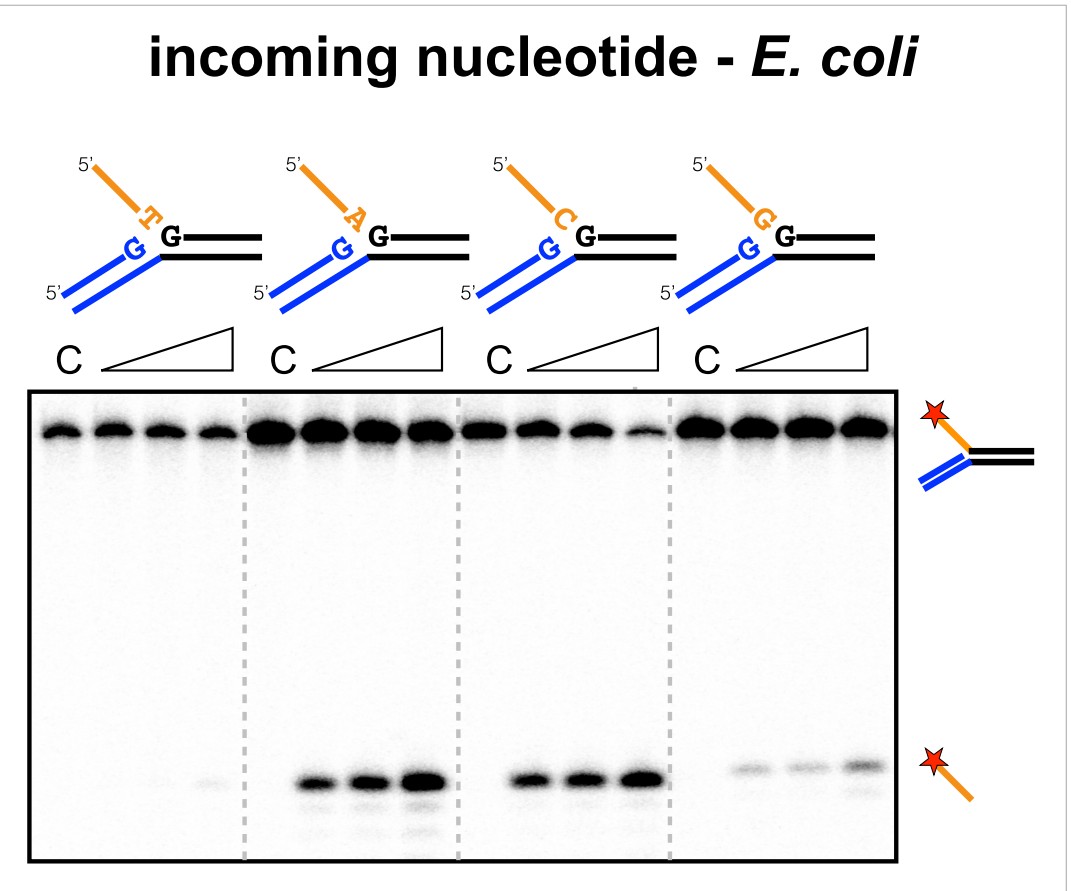

**Figure 8**. Sequence specificity of the EcoCas1 disintegration reaction for the incoming nucleotide. The nucleotide corresponding to the incoming 3′ end of the new spacer, which is the nucleotide at the 3′ end of the 5′-flap in the disintegration substrate, was varied systematically to determine its effect on the disintegration reaction catalysed by EcoCas1 (substrates 2, 3, 4, 5). C–control with no Cas1 added. Time points were 5, 10 and 30 min.

spacer integration. Consequently we still do not have a clear understanding of the roles of Cas1, Cas2 and host proteins in the acquisition mechanism. In this study we investigated the integration reaction by focusing on the biochemistry of the Cas1 protein from *S. solfataricus* and *E. coli*. Efficient TES of branched DNA substrates with a 5′-flap or duplex arm is clearly possible for both *S. solfataricus* and *E. coli* Cas1 in vitro. This is a very precise reaction requiring attack by a 3′-hydroxyl at the branch point, generating a perfect DNA duplex. The reaction almost certainly represents the disintegration reaction that is the reverse of the spacer integration step, as observed for many integrases and transposases where it represents a very useful means to study the underlying integration mechanism (*Gerton et al., 1999*). Evidence for a disintegration activity was recently described by Doudna and colleagues for EcoCas1, but the activity observed was relatively weak, most likely because the branched substrate studied had a non-optimal DNA sequence around the branch point (*Nuñez et al., 2015*).

For the CRISPR adaptation process in vivo, integration occurs at the junction between the first repeat and the leader sequence, which immediately suggests a role for sequence specificity in the reaction. It has also been suggested that CRISPR repeat sequences, which are often palindromic, form four-way DNA junctions in supercoiled DNA, acting as a recognition signal for Cas1, a possibility that is supported by the observation that EcoCas1 can cut four-way DNA junctions in vitro (*Babu et al., 2011*), and the finding that spacer integration in a plasmid lacking a CRISPR locus occurs preferentially next to a palindromic site (*Nuñez et al., 2015*). However, a palindrome alone is not sufficient to support spacer insertion in *E. coli* in vivo (*Arslan et al., 2014*), and this also holds for the type II CRISPR system of *Streptococcus thermophilus* (*Wei et al., 2015*), suggesting that local sequence helps determine the integration site. Furthermore, some CRISPR repeat families, including many in

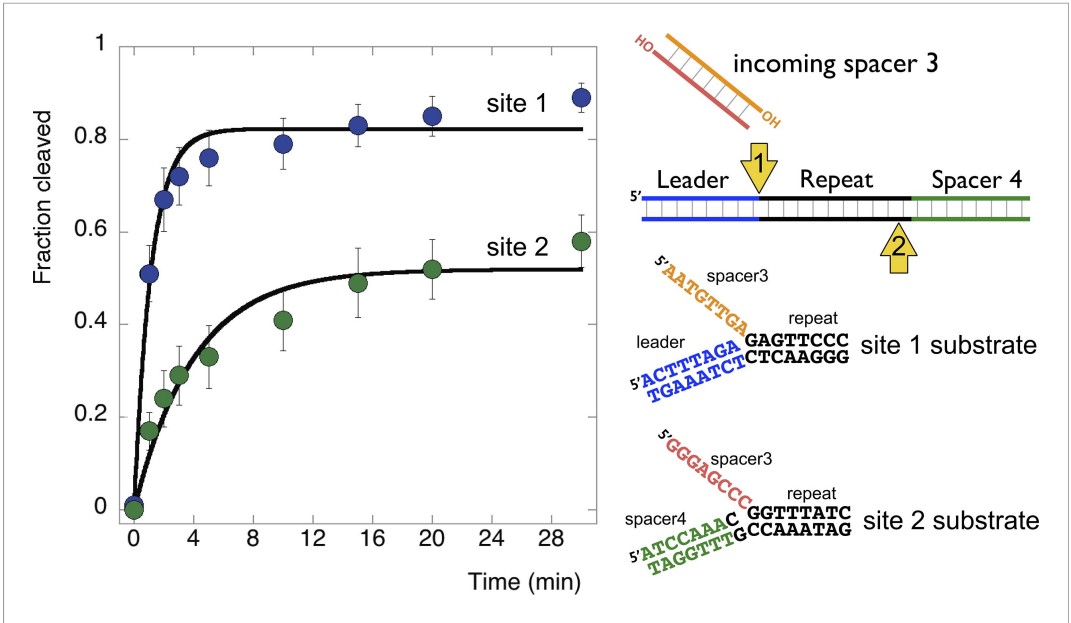

**Figure 9**. Disintegration of authentic *E. coli* integration intermediates. Disintegration substrates corresponding to the expected site 1 and site 2 integration products arising from the integration of spacer 3 into the CRISPR array were constructed and tested (spacer 3-1 and spacer 3-2 substrates). EcoCas1 processed both, with the rate of reaction significantly higher for the substrate corresponding to site 1 (the top strand) at the leader-repeat junction. Data points represent the means of triplicate experiments with standard errors shown (raw data provided in *Figure 9—source data 1*).

The following source data is available for figure 9:

**Source data 1**. *E. coli* Site 1 vs Site 2 time course.

archaea, have little or no palindromic nature and thus cannot form stable hairpin structures (*Kunin et al., 2007*). Cas1 therefore might be expected to act as a sequence specific integrase, although local DNA structure could also play a part.

In support of this hypothesis, both *S. solfataricus* and *E. coli* Cas1 catalyse a disintegration reaction with distinct, sequence specific properties. In particular, there is a clear preference for a guanine at position +1, corresponding to the first nucleotide of the repeat, suggesting that this residue is recognised specifically in the active site of Cas1. The specificity is particularly strong for EcoCas1, consistent with the presence of a guanine in the repeat sequence at the +1 site in both plus and minus strands. Preference for a guanine at the +1 nucleotide for EcoCas1 catalysed integration events has also been observed (*Nuñez et al., 2015*). For SsoCas1, a guanine at this position was preferred over a cytosine, which is the nucleotide present at the +1′ position on the minus strand, by a factor of five. Although the −1 position might also be expected to play a role in the selection of integration sites, deep sequencing data for integration catalysed by EcoCas1 revealed no sequence preference at this position (*Nuñez et al., 2015*). In agreement with this finding, we observed little evidence for sequence discrimination at the −1 position for the disintegration reactions catalysed by either enzyme, with the exception that EcoCas1 disfavours cytosine at this position (*Figure 6B*). A cytosine at position −1 is the expected residue on the minus strand, suggesting that the disintegration reaction may better reflect the reversal of integration at site 1 in the leader-repeat junction. Deep sequencing data for integration reactions catalysed by EcoCas1 in vitro did reveal a marked preference for a guanine at position −2 in the integration site (*Nuñez et al., 2015*). This corresponds well with the −2 position in the plus strand, which is part of the leader sequence and is a guanine in *E. coli*, but cannot hold for the minus strand where the −2′ position is inherently variable in nature. Disintegration of substrates mimicking the integration intermediates relevant for the integration of spacer 3 into the *E. coli* CRISPR array reinforce these conclusions, with site 1 on the plus strand processed significantly more quickly

than site 2 on the bottom strand (*Figure 9*). Taken together, both the disintegration specificity and the deep sequencing data for integration support the hypothesis that integration is targeted to the leader-repeat 1 junction on the plus strand at least in part by the inherent sequence specificity of Cas1, which presumably involves specific recognition of these bases within the active site of the enzyme (*Figure 10*).

For *E. coli* integration reactions in vitro, a marked preference for cytosine over thymine at the 3′ end of the protospacer was observed (*Nuñez et al., 2015*). Furthermore, protospacers with a cytosine at the 3′ end were preferentially incorporated into the minus strand at the junction between repeat 1 and spacer 1. These data are consistent with the requirement for protospacers to supply the final cytosine of the repeat on the minus strand during integration (*Swarts et al., 2012*). The deep sequencing data also revealed a marked preference for adenine over thymine at the 3′ end of the protospacer (*Nuñez et al., 2015*). For disintegration by EcoCas1, we observed a clear preference for adenine or cytosine at the equivalent position, whilst thymine did not support the disintegration reaction (*Figure 8*). Thus EcoCas1 appears to recognise the nucleotide at the 3′ end of the incoming DNA, although no such discrimination was observed for SsoCas1. A dinucleotide sequence ‘AA’ motif over-represented at the 3′ end of protospacers incorporated in *E. coli* strain BL21 has been described previously (*Yosef et al., 2013*).

Although the CRISPR spacer integration system has been compared to the integration and transposition reactions carried out by mobile genetic elements, there is one key difference in the two processes—the length of DNA integrated. The persistence length of DNA, the distance over which it behaves as a fairly rigid rod, is estimated as 35–50 nm (100–150 bp) under conditions found in cells (*Hagerman, 1988*; *Brinkers et al., 2009*). This means that the two ends of a viral genome of several kilobases can be looped around and brought close together relatively easily, but a new spacer of 30–40 base pairs of dsDNA cannot be manipulated in the same way. Considering the scheme in *Figure 1*, the molecular origami required to achieve the second TES reaction looks challenging. Several related enzymes, including Mu transposase (*Savilahti et al., 1995*), V(D)J recombinase (*Ramsden et al., 1996*) and HIV integrase (*Gerton et al., 1999*) are known to disrupt base pairing of DNA substrates and make sequence-specific contacts during the integration reaction. It is likely that Cas1 also manipulates the local DNA duplex structure, which may help in positioning the DNA strands

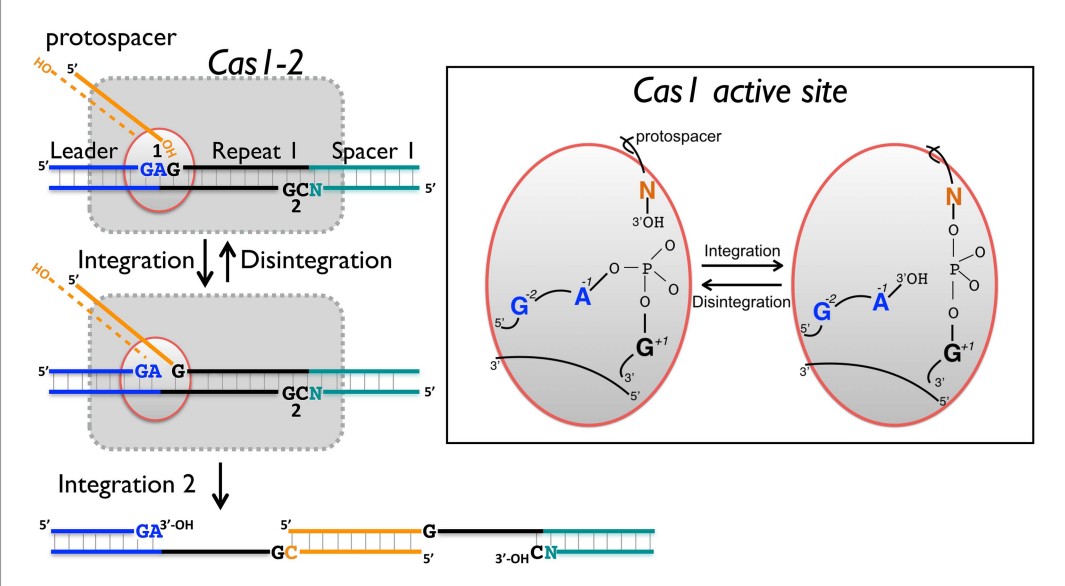

**Figure 10**. Reaction scheme for spacer integration and disintegration by *E. coli* Cas1. The Cas 1-2 complex integrates new spacers via two joining reactions (1 and 2) at either end of the first CRISPR repeat, which differ in their sequence context. Disintegration activity by *E. coli* Cas1 shows clear preference for the sequence at site 1, with the guanines at position +1 and −2 particularly important. At site 2, the sequence context is not optimal for disintegration in vitro, leading to slower reaction rates. In the active site of Cas1, these nucleobases likely make specific interactions with catalytic residues, and the DNA duplex structure may be distorted.

correctly for the TES reaction. The observation that the incoming DNA (the 5′-flap in the disintegration reaction) can be single stranded, partially or fully duplex in nature suggests that there is some flexibility in the recognition of the incoming spacer. This is also consistent with the recent observation of a link between RecBCD, which generates ssDNA products, and Cas1 in *E. coli* (*Levy et al., 2015*). There is no formal requirement that the protospacer should be fully duplex in nature, although current understanding of the integration reaction requires that new spacers have two intact 3′-ends for the two integration reactions so must be at least partially duplex in nature. Many integrases process the ends of the integrated DNA using a nuclease activity, which occurs at the same active site as the integrase activity (*Gerton et al., 1999*). There is no reason to expect that Cas1 will differ in that regard, and indeed the reaction products of the RecBCD nuclease are on average much longer than the DNA molecules integrated by Cas1, suggesting the requirement for further processing.

In conclusion, we have shown that Cas1 from both *S. solfataricus* and *E. coli* have robust TES activities in vitro which reflect the reversal, or disintegration, of the integration reaction. Disintegration is strongly sequence specific, and the specificity fits with the expected sequence for the plus strand at the leader-repeat junction (*Figure 9*). This site is the logical place for the initiation of integration, as it has a unique, defined sequence, in contrast to the repetitive and more variable sequence found at the second integration site on the minus strand. Doudna and colleagues recently proposed a model based on an initial attack at site 2 on the minus strand (*Nuñez et al., 2015*). However, this preference was only significant for spacers with a cytosine at the 3′ terminus, and does not explain the marked preference observed by the authors for a guanine at position +2, which is a feature of the positive strand. Future studies of both the forward and reverse integration reactions catalyzed by Cas1 will help to address these issues and delineate the mechanism of spacer integration in the CRISPR system.

## Materials and methods

### Cloning, gene expression and protein purification

The *sso1450* (*cas1*) gene and *sso1450a* (*cas2*) genes were amplified from *S. solfataricus* P2 genomic DNA by PCR using primer pairs (5′-ATATAACCATGGCAAGCGTGAGGACTT; 5′-TATTGGATCCTCA CATCACCAACTTGAAACCC) and (5′-GCGCCATGGTTACACTAACCATTCCTCTAATC; 5′-GGCCGG ATCCTTGAAATTATTGGTAGTATATGAC), respectively. The amplified genes were cloned into the pEHisTEV vector (*Liu and Naismith, 2009*) downstream of a cleavable His$_6$-tag using *Nco*I and *Bam*HI restriction sites. Site-directed mutagenesis was carried out on the vector containing *sso1450* to mutate active site residue glutamic acid 142 to an alanine using the oligonucleotide sequence 5′-GTTGGATAAGGATGCACCGGCTGCTGCTAG. Standard site-directed mutagenesis protocols (QuikChange, Stratagene, La Jolla, CA, United States) were followed. Mutations were confirmed by sequencing (GATC Biotech, Constance, Germany). The constructs were expressed in C43 (DE3) *E. coli* cells grown in LB (Luria-Bertani) medium supplemented with 35 µg/ml kanamycin to an OD$_{600}$ of 0.6–0.8 at 37°C. Expression was then induced by the addition of 0.4 mM isopropyl-β-D-thiogalactopyranoside (IPTG) and overnight incubation with shaking at 25°C. Cells were harvested (8000×*g*, 20 min) and resuspended in lysis buffer (4.5 mM NaH$_2$PO$_4$, 15 mM Na$_2$HPO$_4$ [pH 7.5], 500 mM NaCl, 30 mM imidazole, 1% Triton-X and protease inhibitors [Roche Applied Science, Basel, Switzerland]). Cells were lysed by sonication, the lysate cleared by ultracentrifugation (90,000×*g*, 35 min) and the supernatant filtered though a 0.22 µm syringe filter and loaded on to a HP HisTrap 5 ml column (GE Healthcare, Little Chalfont, United Kingdom) equilibrated in buffer A (4.5 mM NaH$_2$PO$_4$, 15 mM Na$_2$HPO$_4$ [pH 7.5], 500 mM NaCl, 30 mM imidazole). The His-tagged protein of interest was eluted with a linear gradient from 30 to 500 mM imidazole. Fractions containing Cas1 (or Cas2) were concentrated and buffer exchanged into buffer A, using centrifugal concentrators (30 kDa cutoff, Vivaspin, Sartorius Stedim Biotech GmbH, Goettingen, Germany). His-tags were cleaved by the addition of TEV protease (1:10 ratio of TEV:protein) and overnight incubation at room temperature. The cleaved protein was passed through a HisTrap column in buffer A, and the cleaved protein collected in the flow through. The final purification step was gel filtration on a 26/60 Superdex 200 prep grade column (GE Healthcare) in buffer C (20 mM Tris-HCL (pH 7.5), 500 mM NaCl, 0.5 mM DTT, 1 mM EDTA, 10% glycerol). Purified and concentrated protein samples were flash frozen and stored at −80°C.

Genes encoding *E. coli* K-12 MG1655 Cas1 (*ygbT*) and Cas2 (*ygbF*) were amplified by PCR from genomic DNA using the PCR primer pairs (5′-GGAATTCCATATGACCTGGCTTCCCCTTAATC;

5′-GGAATTCTCAGCTACTCCGATGGCCTGC) and (5′-GGAATTCCATATGAGTATGTTGGTCGTGGT
CAC; 5′-GGAATTCTCAAACAGGTAAAAAAGACAC), respectively. Each PCR product was subcloned
into protein expression plasmid pET14b using restriction enzyme *Nde*I and *Eco*RI. EcoCas1 and Cas2
proteins were over-produced individually in strain BL21 AI (Life Technologies, Carlsbad, CA, United
States), each with N-terminal (His)$_6$ tags. Cells were grown at 37°C to OD$_{600}$ 0.5–0.6 in LB broth
containing ampicillin (50 μg/ml) and induced using IPTG (1 mM) and arabinose (0.2% wt/vol), with
growth continued for 3 hr after induction. Cas1 or Cas2 expressing cells were harvested for re-
suspension in buffer H (20 mM Tris.HCl pH7.5, 500 mM NaCl, 5 mM imidazole, 10% glycerol) and flash
frozen in liquid nitrogen for storage at −80°C until required. The first purification step was identical for
both Cas1 and Cas2: sonicated biomass was clarified by centrifugation (90,000×*g*, 25 min) and soluble
extract was passed into a 5 ml Hi-Trap Nickel chelating column (GE Healthcare) equilibrated with
buffer H. Cas1 or Cas2 eluted at 70–100 mM imidazole in a linear imidazole gradient. Sodium chloride
was reduced to 50 mM by dialysis against buffer S (20 mM Tris.HCl pH7.5, 50 mM NaCl, 1 mM DTT,
10% glycerol). Cas1 was loaded onto a 5 ml Hi-Trap heparin column and eluted in a gradient of NaCl
at 200–300 mM in buffer S. Pooled Cas1 fractions were loaded directly onto a S300 size exclusion
column equilibrated in buffer S with 250 mM NaCl. Cas1 fractions were pooled for storage at −80°C in
buffer S containing 250 mM NaCl and 40% glycerol. Desalted Cas2 eluted from Ni-NTA was dialyzed
into buffer S containing 1.5 M NaCl and applied to a 5 ml Hi-Trap butyl-Sepharose column (GE
Healthcare), eluting in the flow through. Cas2 fractions were pooled and loaded directly onto a S300
size exclusion column equilibrated in buffer S with 250 mM NaCl. Following isocratic elution, Cas2
fractions were pooled and stored as for Cas1.

## Sequence and preparation of DNA substrates

Substrates were purchased from Integrated DNA Technologies (Coralville, IA, United States) either
unlabeled or with a 3′-fluorescein label (*Table 1*). Oligonucleotides were 5′-$^{32}$P-radiolabelled and gel
purified as described previously (*Hutton et al., 2010*). Labelled oligonucleotides were annealed with
complementary strands by heating with an excess of unlabelled strands at 95°C for 5 min and then slow
cooling to room temperature overnight in a heating block. The assembled substrates (*Table 2*) were purified
by native polyacrylamide (12%) gel electrophoresis with 1× Tris-borate-EDTA (TBE) buffer, followed by band
excision, gel extraction and ethanol precipitation before being resuspended in nuclease free water, as
described previously (*Hutton et al., 2010*). The final substrate concentration was measured using the
extinction coefficient and absorbance at 260 nm in a UV-Vis spectrophotometer (Varian Cary, Agilent, Santa
Clara, CA, United States) and DNA diluted to ~50 nM or ~200 nM final concentration for use in assays.

## Disintegration reactions

Reactions were typically carried out under single turnover conditions. Titration of SsoCas1 (*Figure 3B,C*)
showed evidence for inhibition at enzyme:substrate ratios above 10:1. For standard assays, 2 μM Cas1
protein was mixed with 200 nM substrate in cleavage buffer (20 mM Tris [pH 7.5], 10 mM NaCl, 1 mM
DTT and 5 mM MnCl$_2$) and incubated at 55°C (for SsoCas1) or 37°C (for EcoCas1). For reactions with
Cas1 and Cas2, the proteins were mixed in equimolar concentration and incubated together at either
37 or 55°C for 30 min before being added to the reaction. At indicated times, reactions were quenched
by the addition of EDTA to 20 mM final concentration and 1 μl 20 mg/ml Proteinase K (Promega,
Madison, WI, United States) and incubation at 37°C for 30 min. The DNA was then separated from the
reaction by phenol chloroform extraction. 60 μl neutral phenol:chloroform:isoamyl alcohol (Sigma–
Aldrich, St. Louis, MO, United States) was added and the reaction vortexed for 30 s. The sample was
then centrifuged (15,000×*g*, 1 min) and the upper aqueous phase, containing the DNA, collected.
Formamide loading dye (100% formamide with 0.25% bromophenol blue, 0.25% xylene cyanol) was
added (5 μl) and the sample heated at 95°C for 2 min before being chilled on ice. Reaction products were
resolved on a pre-run 20% denaturing (7 M urea) polyacrylamide gel containing 1× TBE in 1× TBE buffer.
Gels were run at 80 W, 45°C for 90 min before overnight exposure to a phosphorimaging plate and
imaging with a FLA-5000 Imaging System (Fujifilm Life Science, Düsseldorf, Germany).

## *Sac*I site repair

Assays with the *Sac*I junction substrate were carried out under standard conditions with SsoCas1 for
30 min. FastDigest *Sac*I enzyme (1 μl) and 1 μl FastDigest Buffer (Thermo Scientific, Waltham, MA,

**Table 1.** Sequence of oligonucleotides used for substrate construction

| Strand | Sequence 5'→3' | Length |
|---|---|---|
| 1a | TAGTAAGAGATTAATAAACCCTCAGATAATCTCTTATAGAATTGAAAGTTCGG | 53 |
| 1b | TTTTTTTTTTTTTTTTTATTATCTGAGGGTTTATTAATCTCTTACTA | 48 |
| 1c | CCGAACTTTCAATTCTATAAGAG | 23 |
| 2a | TAGTAAGAGATTAATAAACCCTCAGATAACCTCTTATAGAATTGAAAGTTCGG | 53 |
| 2b | TTTTTTTTTTTTTTTTTGTTATCTGAGGGTTTATTAATCTCTTACTA | 48 |
| 3b | TTTTTTTTTTTTTTTTTAGTTATCTGAGGGTTTATTAATCTCTTACTA | 48 |
| 4b | TTTTTTTTTTTTTTTTTCGTTATCTGAGGGTTTATTAATCTCTTACTA | 48 |
| 5b | TTTTTTTTTTTTTTTTTGGTTATCTGAGGGTTTATTAATCTCTTACTA | 48 |
| 6a | TAGTAAGAGATTAATAAACCCTCAGATAAGCTCTTATAGAATTGAAAGTTCGG | 53 |
| 6b | TTTTTTTTTTTTTTTTTACTTATCTGAGGGTTTATTAATCTCTTACTA | 48 |
| 7a | TAGTAAGAGATTAATAAACCCTCAGATAAACTCTTATAGAATTGAAAGTTCGG | 53 |
| 7b | TTTTTTTTTTTTTTTTTATTTATCTGAGGGTTTATTAATCTCTTACTA | 48 |
| 8b | TTTTTTTTTTTTTTTTTAATTATCTGAGGGTTTATTAATCTCTTACTA | 48 |
| 9a | TAGTAAGAGATTAATAAACCCTCAGATAACATCTTATAGAATTGAAAGTTCGG | 53 |
| 9c | CCGAACTTTCAATTCTATAAGAT | 23 |
| 10a | TAGTAAGAGATTAATAAACCCTCAGATAACTTCTTATAGAATTGAAAGTTCGG | 53 |
| 10c | CCGAACTTTCAATTCTATAAGAA | 23 |
| 11a | TAGTAAGAGATTAATAAACCCTCAGATAACGTCTTATAGAATTGAAAGTTCGG | 53 |
| 11c | CCGAACTTTCAATTCTATAAGAC | 23 |
| 12a | TAGTAAGAGATTAATAAACCCTCAGATAACTCCTTATAGAATTGAAAGTTCGG | 53 |
| 12c | CCGAACTTTCAATTCTATAAGGA | 23 |
| 13a | TAGTAAGAGATTAATAAACCCTCAGATAACTGCTTATAGAATTGAAAGTTCGG | 53 |
| 13c | CCGAACTTTCAATTCTATAAGCA | 23 |
| 14a | TAGTAAGAGATTAATAAACCCTCAGATAACTACTTATAGAATTGAAAGTTCGG | 53 |
| 14c | CCGAACTTTCAATTCTATAAGTA | 23 |
| SacI-a | TAGTAAGAGATTAATAAACCCTCAGATGAGCTCTTATAGAATTGAAAGTTCGG | 53 |
| SacI-b | TTTTTTTTTTTTTTCTCATCTGAGGGTTTATTAATCTCTTACTA | 44 |
| 1b-3'-FAM | TTTTTTTTTTTTTTTTTATTATCTGAGGGTTTATTAATCTCTTACTA-FAM | 48 |
| 19a | CCTCGAGGGATCCGTCCTAGCAAGCCGCTGCTACCGGAAGCTTCTGGACC | 50 |
| 19b | GCTCGAGTCTAGACTGCAGTTGAGAGCTTGCTAGGACGGATCCCTCGAGG | 50 |
| 19c | GGTCCAGAAGCTTCCGGTAGCAGCG | 25 |
| 20d-10 | AGTCTAGACTCGAGC | 15 |
| 20d-5 | ACTGCAGTCTAGACTCGAGC | 20 |
| 20d | TCTCAACTGCAGTCTAGACTCGAGC | 25 |
| 25c-d | GGTCCAGAAGCTTCCGGTAGCAGCGTCTCAACTGCAGTCTAGACTCGAGC | 50 |
| 1c-3'P | CCGAACTTTCAATTCTATAAGAG-phos | 25 |
| Sp3-1a | CTGGCGCGGGGAACTCTCTAAAAGTATACATTTGTTCTT | 39 |
| Sp3-1b | TGTAATTGATAATGTTGAGAGTTCCCCGCGCCAG | 34 |
| Sp3-1c | AAGAACAAATGTATACTTTTAGA | 23 |
| Sp3-2a | CCAGCGGGGATAAACCGTTTGGATCGGGTCTGGAATTTC | 39 |
| Sp3-2b | TGTTCCGACAGGGAGCCCGGTTTATCCCCGCTGG | 34 |
| Sp3-2c | GAAATTCCAGACCCGATCCAAAC | 23 |

**Table 2.** DNA constructs used in this study

| Substrate | Oligonucleotide components | Junction sequence | | | |
|---|---|---|---|---|---|
| | | −2 | −1 | 1 | IC |
| Substrate 1 | 1a, 1b, 1c | A | G | A | T |
| Substrate 1-FAM | 1a, 1b-3'-FAM, 1c | A | G | A | T |
| SacI substrate | SacI-a, SacI-b, 1c | A | G | C | T |
| Substrate 2 | 2a, 2b, 1c | A | G | G | T |
| Substrate 3 | 2a, 3b, 1c | A | G | G | A |
| 3'-phos substrate | 2a, 3b, 1c-3'P | A | G | G | A |
| Substrate 4 | 2a, 4b, 1c | A | G | G | C |
| Substrate 5 | 2a, 5b, 1c | A | G | G | G |
| Substrate 6 | 6a, 6b, 1c | A | G | C | A |
| Substrate 7 | 7a, 7b, 1c | A | G | T | A |
| Substrate 8 | 1a, 8b, 1c | A | G | A | A |
| Substrate 9 | 9a, 3b, 9c | A | T | G | A |
| Substrate 10 | 10a, 3b, 10c | A | A | G | A |
| Substrate 11 | 11a, 3b, 11c | A | C | G | A |
| Substrate 12 | 12a, 3b, 12c | G | A | G | A |
| Substrate 13 | 13a, 3b, 13c | C | A | G | A |
| Substrate 14 | 14a, 3b, 14c | T | A | G | A |
| Substrate 15 | 2a, 4b, 1c | A | G | G | C |
| Substrate 16 | 11a, 4b, 11c | A | C | G | C |
| Substrate 17 | 10a, 4b, 10c | A | A | G | C |
| Substrate 18 | 9a, 4b, 9c | A | T | G | C |
| Substrate 19 | 19a, 19b, 19c | C | G | G | A |
| Gap10 | 19a, 19b, 19c, 20d-10 | C | G | G | A |
| Gap5 | 19a, 19b, 19c, 20d-5 | C | G | G | A |
| Nick | 19a, 19b, 19c, 20d | C | G | G | A |
| Y-junction | 19a, 19b, 20c-d | C | G | G | A |
| Spacer 3-1 substrate | Sp3-1a, Sp3-1b, Sp3-1c | G | A | G | A |
| Spacer 3-2 substrate | Sp3-2a, Sp3-2b, Sp3-2c | A | C | G | C |

The sequence of the central portion of the junction (positions −2, −1, 1 and the incoming nucleotide (IC)) for each substrate is shown. The oligonucleotides used to assemble the complete substrate are indicated.

United States) were then added and the reaction incubated at 37°C for 30 min. Product extraction, separation and visualization was then carried out as described above.

## Disintegration reaction time courses

For the time course assays, the concentration of DNA substrates was 50 nM and the concentration of Cas1 protein 50 nM for SsoCas1 or 500 nM for EcoCas1. Reactions were performed as described above with the omission of the Proteinase K step. Following phosphorimaging, substrates and products were quantified using Image Gauge software (Fujifilm) and the reaction course was plotted using Kaleidagraph (Synergy Software, Reading, PA, United States). Experiments were carried out in triplicate and the mean and standard error calculated for each point. For SsoCas1, the data were fitted using a single exponential (*Niewoehner et al., 2014*). For EcoCas1 the reactions did not go to completion and were therefore fitted with a floating end-point, as described previously (*Niewoehner et al., 2014*).

## Acknowledgements

This work was supported by a grant from the Biotechnology and Biological Sciences Research Council (REF: BB/M000400/1 to MFW). Thanks to Shirley Graham and Kotryna Temcinaite for technical support, and Jing Zhang and Agnes Tello for helpful discussions.

## Additional information

### Funding

| Funder | Grant reference | Author |
|---|---|---|
| Biotechnology and Biological Sciences Research Council (BBSRC) | BB/M000400/1 | Malcolm F White |

The funder had no role in study design, data collection and interpretation, or the decision to submit the work for publication.

### Author contributions

CR, Conception and design, Acquisition of data, Analysis and interpretation of data, Drafting or revising the article; SS, Acquisition of data; ASB, Acquisition of data, Contributed unpublished essential data or reagents; ELB, Drafting or revising the article, Contributed unpublished essential data or reagents; MFW, Conception and design, Analysis and interpretation of data, Drafting or revising the article

### Author ORCIDs

Malcolm F White, http://orcid.org/0000-0003-1543-9342

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
