## [Decision Letter]

Thank you for submitting your work entitled “Intrinsic sequence specificity of the Cas1 integrase directs new spacer acquisition” for peer review at *eLife*. Your submission has been favorably evaluated by Jim Kadonaga (Senior Editor), Timothy Nilsen (Reviewing Editor), and three reviewers, one of whom, Erik Sontheimer, has agreed to share his identity.

We have received comments on your manuscript from three experts in the CRISPR field. After discussion among them, it was agreed that you describe work that significantly advances this area. Please address the rather minor comments of Reviewers 1 and 2 and perform the experiments requested by Reviewer 3.

Reviewer #1:

Rollie et al. present a series of studies on Cas1 (and, to a lesser extent, Cas2) from both *E. coli* and *S. solfataricus*, and on their activities in vitro that relate to their cellular roles in CRISPR spacer acquisition. The authors define an activity that is an apparent reversal (disintegration) of the natural integration pathway, with a 5'-flap substrate undergoing a transesterification reaction that releases that single-stranded 5'-flap in concert with the joining of the two flanking base-paired segments. This reaction has previously been used to study the activities of other integrases and recombinases, and also of *E. coli* Cas1/Cas2 in the work of Nunez et al. (Nature, 2015). The appearance of the latter study partly stole a march on the current work, with the primary contributions of this manuscript (i.e. above and beyond the results presented by Nunez et al.) being the inclusion of a separate, distinct Cas1 (SsoCas1) and the more detailed analysis of the sequence preferences of the reactions catalyzed by both Eco and SsoCas1. These contributions are significant, in part because the leader and repeat DNA sequences (where the reactions occur) differ between the relevant strains of *E. coli* and *S. solfataricus*, allowing comparative tests of substrate specificity, and of the roles of Cas1 in establishing that specificity. The authors do in fact establish such differences, and overall the reaction data that is presented is compelling.

A critical question is the degree to which the requirements for disintegration reflect those of the presumptive integration. As an example, the initial protospacer attack (forward) reaction described by Nunez et al. requires a double-stranded protospacer, whereas the disintegration liberates single-stranded (as well as double-stranded) flaps; similarly, the inclusion of Cas2 is far more important for integration than for disintegration. Overall, the history of integrase/transposase research suggests that the reverse reaction can be a useful and convenient proxy for the forward reaction, but this nonetheless calls for some caution in interpretation.

One other potential concern is that the authors used enzyme concentrations that were rather high – 2 micromolar, according to the Materials and methods section. This is more than twenty-five-fold higher than those used by Nunez et al. Is there a reason for this? The higher concentrations could drive enzyme/substrate interactions that might not occur at concentrations that are probably closer to physiological, and could therefore obviously bear on questions of specificity. This concern is particularly meaningful in light of Figure 3, which shows good activity in the low- to mid-nanomolar range. In fact, in light of the SsoCas1 data in Figure 3, 2-micromolar reactions should not work at all for that protein. Can the authors explain this discrepancy?

Perhaps the most meaningful question is about the degree to which the results described in this manuscript move the field forward, relative to expectations for this journal. My initial reaction is that the differences in substrate specificity, and the roles of these divergent Cas1 orthologs in conferring that specificity, represent a reasonably significant advance, though this is a somewhat close call.

Minor comments:

It could be useful for the authors to tell the reader the CRISPR-Cas subtypes from which these particular Cas1/Cas2 orthologs were derived.

Reviewer #2:

Genetic studies have shown that Cas1 is critical for novel spacer integration at CRISPR loci. Recent work with recombinant *E. coli* Cas1 by the Doudna group (Nunez et al., Nature Structural Biology, 2014, 21:528-534 and Nunez et al., Nature, 2015, 519:193-198) provided strong evidence that Cas1 functions as the integrase and that its dual transeserification activity displays sequence specificity and is strongly dependent upon association with the Cas2 protein. Furthermore, Nunez et al. demonstrated that 33 bp double-stranded DNA fragments can be integrated into CRISPR repeats by the Cas1 and Cas2 proteins and both the forward (integration) and reverse (disintegration) reactions were observed and characterized. Here, Rollie et al. report that Cas1 alone (from *E. coli* or *S. solfataricus* sources) can catalyze a single transesterification reaction on 5' flap DNA substrates that possibly (but not definitively) mimics a disintegration reaction. The main claim in this paper is that Cas1 shows a clear sequence preference for nucleotides at the leader-repeat boundary but this is not true for both sites of integration.

The current study suffers in that it does not significantly expand our mechanistic understanding of Cas1 function or spacer acquisition beyond that reported by others ([24]; Babu et al., Mol. Micro., 2011, 79:484-502). Indeed, much of the analysis reaches the conclusion that Cas1 can carry out transesterification reactions on certain branched DNA substrates–the novelty of this observation is undermined, since a transesterification mechanism was already clearly demonstrated by the recent work by Doudna and colleagues (24). Another weakness of the paper is that it examines only the putative reverse (disintegration) reaction and the validity of the conclusion that the reaction being studied is reflective of the integration event must await reconstitution of the forward (integration) reaction. Moreover, the claim of clear sequence specificity that coincides with the in vivo integration site is not clear from the data shown. The order of sequential transesterifications resulting in spacer acquisition differs between this work and those of [24]. Indeed, by systematically varying the nucleotides flanking each transesterification site, it was concluded that there is no obvious sequence specificity for site 2 (Figure 6).

Curiously, site 2 described in the paper is equivalent to the site 1 of [24], which was the main site used where sequence-specificity was observed and which accounts for the in vivo observation that newly acquired spacers have a 5' G as the first nt. flanking the leader-repeat junction that is derived from the upstream PAM. In general, the data in the paper is of high quality and the observed nucleotide specificity at +1 position of site 1 seems solid. However, it is unclear if the reaction under investigation provides a strong model for learning how the spacer acquisition reaction occurs in vivo.

Minor comments:

A relevant reference showing that nucleotides at the leader-repeat boundary are critical for new spacer acquisition was omitted and should be cited at the appropriate locations within the Introduction and Discussion. The reference is: Wei et al., Nucleic Acids Res., 2015, 43:1749-58.

Reviewer #3:

Rollie and co-workers have characterized the sequence specificity of Cas1 from *E. coli* and *S. solfataricus*, which has been shown to be involved in the integration of spacers in the CRISPR memory locus. The authors show that these enzymes display efficient transesterification reactions, which may be considered the reversal of the integration of spacers. The manuscript is interesting, important, well written, and biochemically thorough, but would benefit from some improvements described below.

One key experiment seems to be missing. It remains unclear whether EcoCas1 prefers the disintegration reactions at site 1 or site 2. Although the authors state that the disintegration reaction occurs at site 2 as well, the design of the substrates for the disintegration reaction seems to be based on site 1, which is the leader-repeat boundary. Because a Cas1-Cas2 complex has recently been shown to start integrating spacers from site 2 (24), it would be good to directly compare the kinetics of both disintegration reactions using the exact substrates that are now thought to be representing site 1 (leader - repeat), and other substrates representing site 2 (repeat - spacer 1), for both EcoCas1 and EcoCas1-Cas2. Has this been done? This will help understand the order of events leading to spacer integration, and take this paper to a next level.

Minor comments:

1) The model in the left panel of Figure 9 could be expanded to include more details (e.g. numbering of positions) and more steps, even if they are hypothetical. This will help understand how full spacer integration is achieved and would increase the readability of the paper.

2) It would be good to make the figures as intuitive as possible. This includes changing lane numbering and legend description (e.g. in Figure 2) into lane text labeling in the figure.

3) Can the markers used to indicate nucleobases C, T and A in Figures 5 and 7 be changed to something that is better visible (solid circle, square etc)?

---

## [Author Response]

Reviewer #1:

*[…] A critical question is the degree to which the requirements for disintegration reflect those of the presumptive integration. As an example, the initial protospacer attack (forward) reaction described by Nunez et al. requires a double-stranded protospacer, whereas the disintegration liberates single-stranded (as well as double-stranded) flaps; similarly, the inclusion of Cas2 is far more important for integration than for disintegration. Overall, the history of integrase/transposase research suggests that the reverse reaction* can *be a useful and convenient proxy for the forward reaction, but this nonetheless calls for some caution in interpretation*.

*One other potential concern is that the authors used enzyme concentrations that were rather high – 2 micromolar, according to the Materials and methods section. This is more than twenty-five-fold higher than those used by Nunez et al. Is there a reason for this? The higher concentrations could drive enzyme/substrate interactions that might not occur at concentrations that are probably closer to physiological, and could therefore obviously bear on questions of specificity. This concern is particularly meaningful in light of*
Figure 3*, which shows good activity in the low- to mid-nanomolar range. In fact, in light of the SsoCas1 data in*
Figure 3, *2-micromolar reactions should not work at all for that protein. Can the authors explain this discrepancy?*

Nuclease activity is typically measured under single turnover conditions with high enzyme excess. We show that disintegration activity can be observed over a wide range of Cas1 concentrations (Figure 3) and have chosen concentrations that give a robust activity and therefore signal. The apparent discrepancy for the 2 µM SsoCas1 reactions is explained by the fact that the inhibition observed at high concentrations for SsoCas1 is dependent on the ratio of substrate DNA to protein. The reactions with 2 µM SsoCas1 used 200 nM substrate, whereas in Figure 3 substrate concentration is 50 nM. This has now been explained in the Methods.

Perhaps the most meaningful question is about the degree to which the results described in this manuscript move the field forward, relative to expectations for this journal. My initial reaction is that the differences in substrate specificity, and the roles of these divergent Cas1 orthologs in conferring that specificity, represent a reasonably significant advance, though this is a somewhat close call.

*Minor comments*:

*It could be useful for the authors to tell the reader the CRISPR-Cas subtypes from which these particular Cas1/Cas2 orthologs were derived*.

This has now been added to the beginning of the Results.

Reviewer #2 (Minor comments):

*A relevant reference showing that nucleotides at the leader-repeat boundary are critical for new spacer acquisition was omitted and should be cited at the appropriate locations within the Introduction and Discussion. The reference is: Wei et al*.*, Nucleic Acids Res., 2015, 43:1749-58.*

This reference was already cited at the appropriate place in the Discussion.

Reviewer #3:

*[…] One key experiment seems to be missing. It remains unclear whether EcoCas1 prefers the disintegration reactions at site 1 or site 2. Although the authors state that the disintegration reaction occurs at site 2 as well, the design of the substrates for the disintegration reaction seems to be based on site 1, which is the leader-repeat boundary. Because a Cas1-Cas2 complex has recently been shown to start integrating spacers from site 2 (*[24]*), it would be good to directly compare the kinetics of both disintegration reactions using the exact substrates that are now thought to be representing site 1 (leader - repeat), and other substrates representing site 2 (repeat - spacer 1), for both EcoCas1 and EcoCas1-Cas2. Has this been done? This will help understand the order of events leading to spacer integration, and take this paper to a next level*.

This is an excellent suggestion. We have constructed DNA substrates that correspond to the site 1 and site 2 integration products for the addition of spacer 3 into the CRISPR array (new Figure 9). The disintegration activity of EcoCas1 with these substrates has been quantified in triplicate and plotted. Both substrates support disintegration, with the activity at site 1 significantly higher than for site 2. This reinforces our conclusion that disintegration can be viewed as a reversal of the half–integration reaction and that specificity of EcoCas1 for site 1 appears higher than for site 2.

*Minor comments*:

*1) The model in the left panel of*
Figure 9
*could be expanded to include more details (e.g. numbering of positions) and more steps, even if they are hypothetical. This will help understand how full spacer integration is achieved and would increase the readability of the paper*.

We agree. We have added the second integration step to this model (now Figure 10) and expanded the legend to explain this in more detail. We agree this improves the clarity.

*2) It would be good to make the figures as intuitive as possible. This includes changing lane numbering and legend description (e.g. in*
Figure 2*) into lane text labeling in the figure*.

We have labelled each lane in Figure 2 to aid the reader as suggested.

*3) Can the markers used to indicate nucleobases C*, *T and A in*
Figures 5 and 7
*be changed to something that is better visible (solid circle, square etc)?*

These have now been redesigned to aid visibility.